# The Biocompatibility of a New Erythritol-and Xyltol-Containing Fluoride Toothpaste

**DOI:** 10.3390/healthcare9080935

**Published:** 2021-07-25

**Authors:** Barbara Cvikl, Adrian Lussi

**Affiliations:** 1Department of Conservative Dentistry, Sigmund Freud University, A-1020 Vienna, Austria; 2Center for Dental Medicine, Department of Operative Dentistry and Periodontology, DE-79106 Freiburg im Breisgau, Germany; adrian.lussi@zmk.unibe.ch

**Keywords:** biocompatibility, erythritol, LC50, toothpaste

## Abstract

The basic function of toothpastes is biofilm removal in order to prevent caries and gingivitis. Toothpastes should provide maximal fluoride availability, optimal abrasivity, and ingredients that do not interfere with fluoride release but should have additional beneficial effects. Further, the effect on cells of the oral cavity is of the utmost importance. We investigated several biological parameters of a new toothpaste (AirFlow-AF) that contains fluoride, xylitol and erythritol but no sodium lauryl sulfate and compared them to commercially available toothpastes (Zendium-Ze, Sensodyne-Se, OdolMed-OM, OralB-OB). The half lethal concentration (LC50) as well as the proliferation behavior on gingival (GF), periodontal ligament (PDL), and mouse fibroblast cells (L929) were was tested. The mean LC50 values of AF on GF, PDL, and L929 were 16.2, 10.9, and 9.3, respectively. In comparison, the four other toothpastes showed mean LC50 values of 1.5 (OB), 1.2 (OM), 1.4 (Se), and 27.7 (Ze) on GF. Mean LC50 values on PDL and L929 were 1.0 and 0.2 (OB), 3.7 and 0.9 (OM), 1.2 and 0.6 (Se), and 25.4 and 5.6 (Ze), respectively. Proliferation behavior mainly confirmed the LC50 values. While cells after stimulation with AF returned to almost unimpaired proliferation behavior at 6%, cells were still strongly impaired after stimulation with all tested commercially toothpastes. AF showed high biocompatibility with different cell types.

## 1. Introduction

Caries and periodontal diseases still exist, negatively influencing both individual well-being and the general economy [1]. Although the fight against these diseases seems very simple—daily removal of the dental plaque using fluoridated toothpaste—there is no consensus on which product and which specific ingredients are recommended. In addition to the actual requirements that a toothpaste has to fulfil, such as cleaning the teeth, releasing fluoride, and others [2], a toothpaste must also be well tolerated by the tissue surrounding the teeth. Basic ingredients of modern toothpastes are abrasives, surfactants, viscosity and rheology modifiers, humectants, flavors, sweeteners, colorants, preservatives, and water. Furthermore, numerous active ingredients are added [2]. The most important ingredient in varying concentrations depending on the age of the user is undoubtedly fluoride. The use of fluoride in daily oral hygiene has led to a significant decrease in caries and can certainly be considered a milestone in modern dentistry.

In addition to fluoride, there are other naturally occurring substances that may be effective against the risk of tooth decay. Examples of these are erythritol and xylitol. Erythritol is a naturally occurring sugar alcohol that is approved as a sugar substitute with no or minimal laxative effect and without any quantity restriction as a food additive. In various in vitro and clinical studies, erythritol showed an inhibitory effect on biofilm formation, on the presence of caries-inducing bacteria, and on the susceptibility of subjects to caries [3,4,5]. Xylitol, also a naturally occurring sugar alcohol, is currently used mainly as a sugar substitute because it has no or minimal acidogenic potential. In addition to the saliva-promoting effect, when using a xylitol-containing chewing gum or lozenge, there is an inhibitory effect on the formation of plaque. In addition, Streptococcus mutans, one of the leading germs in caries formation, cannot successfully metabolize xylitol, which reduces the volume of plaque and thus the production of extracellular polysaccharides. Even xylitol-resistant Streptococcus mutans strains, which have been shown to develop with frequent contact with the sugar substitute, are less cariogenic because they adhere less strongly to the tooth structure, making the biofilm more labile [6,7]. Thus, although erythritol and xylitol do not possess the same caries-protective effect as fluoride, they may be beneficial as additives.

In contrast to fluoride, sodium lauryl sulfate (SLS) is an ingredient that should be viewed with some caution. SLS belongs to the group of surfactants and its foaming effect ensures good distribution of the toothpaste in the oral cavity. In addition, surfactants reduce the surface tension due to their simultaneous hydrophobic and hydrophilic properties, which gives them a cleansing effect. Although SLS is one of the less expensive surfactants and is still widely used, some toothpaste manufacturers are already abandoning SLS as a surfactant because it is considered problematic for the cells of the oral mucosa in some in vitro studies. Furthermore, clinical reports on the association between more frequent occurrence and longer persistence of ulcerative aphthae and the use of SLS-containing toothpastes reinforce the use of alternative surfactants [8,9,10].

What we already know is that toothpastes that contain certain ingredients for foaming have negative effects on cells in vitro and also on the oral mucosa in vivo [11,12,13]. For this reason, a different detergent was used in the new toothpaste to be investigated. Moreover, in a possible synergistic effect on the prevention of caries and plaque, erythritol and xylitol were added to the fluoridated toothpaste. The aim of the present study was to compare the newly formulated AirFlow toothpaste with SLS-containing toothpastes and one SLS-free toothpaste in terms of its safety for oral mucosal tissues using cell experiments.

## 2. Materials and Methods

### 2.1. Toothpaste Slurries for Cell Incubation Procedures

The toothpastes used for the experiments were AirFlow-AF (Dr. Wittmann GmbH & Co.KG, Zwingenberg, Germany), Zendium Complete Protection-Ze (Unilever, Hamburg, Germany), Sensodyne Repair-Se, Odol Med3 Classic-OM, and Oral-B Repair-OB (all three GlaxoSmithKline, Brentford, UK). Information on the amount of fluoride added and the surfactant used is summarized in Table 1. Toothpaste slurries for cell incubation procedures were gained as follows: toothpastes were diluted in serum-free medium (50 *w*/*v*%) and dissolved in sealable plastic tubes with a magnetic stirrer bar at 350 rpm. Afterwards, centrifugation at 16,000× *g* for 10 minutes of the slurries was performed and the toothpaste-conditioned medium (TCM) was filter-sterilized [11,12]. For the individual experiments, the TCM was diluted in a 1:2 dilution series to a final concentration of 0.8 *w*/*v*% toothpaste in serum-free medium.

#### 2.1.1. Measurement of the pH Value and Free Fluoride

The pH of each toothpaste was measured at room temperature using a pH meter (691 pH Meter, Metrohm, Switzerland) after mixing 20 g toothpaste and 40 g artificial saliva with an IKA Ultra-Turrax T 25 digital Dispenser for 10 s.

The free fluoride was measured using a fluoride ion-specific electrode (Mettler Toledo, Scherzenbach, Switzerland). To do so, paste slurries as above were prepared and homogenized by agitation for 1 h and the fluoride ion concentration was calculated and expressed in parts per million (ppm). The calibration curve in the expected fluoride range was performed. All measurements were made in triplicate.

#### 2.1.2. Cell Culture

A murine cell line (L929) often used to perform cytotoxicity tests and human cells of the oral cavity exposed to toothpaste during oral hygiene (gingival fibroblasts—GF and periodontal ligament cells—PDL) were used to evaluate the half lethal concentration of the toothpastes. Furthermore, the proliferation behavior of the cells after incubation with the corresponding toothpaste slurries was investigated. Human gingiva and periodontal ligament fibroblasts were obtained from tissue grafts after approval by the local ethics committee (123/2020 SFU) and after informed consent by the donors. To isolate the cells, tissue explants from the periodont and the gingiva of extracted wisdom teeth were cultivated in Dulbecco’s Modified Eagle Medium (DMEM, Invitrogen Corporation, Carlsbad, CA, USA) supplemented with 10% fetal calf serum (FCS; PAA Laboratories, Linz, Austria) and antibiotics (Invitrogen) at 37 °C, 5% CO_2_ and 95% humidity. Fibroblasts that grew out from the explants were used for the experiments up to the seventh passage. For the respective experiments, the three types of cells were each seeded in microtiter plates (Greiner Bio-One GmbH, Frickenhausen, Germany) at 30,000 cells/cm^2^ one day before incubation with the different toothpastes.

#### 2.1.3. Half Lethal Concentration (LC50)

L929 cells, gingival fibroblasts (GF), and periodontal ligament cells (PDL), all seeded in growth medium at 30,000 cells/cm^2^ into 96-well plate culture dishes (Greiner Bio-One GmbH, Frickenhausen, Germany), were incubated with different concentrations (50 *w*/*v*%, 25 *w*/*v*%, 12.5 *w*/*v*%, 6.3 *w*/*v*%, 3.1 *w*/*v*%, 1.6 *w*/*v*%, and 0.8 *w*/*v*%) of freshly prepared TCM. After two minutes, cells were washed with phosphate-buffered saline (PBS), which was replaced by serum-free media containing MTT (3-[4,5-dimethythiazol-2-yl]-2,5-diphenyltetrazolium bromide, 0.5 mg/mL, Sigma-Aldrich, St. Louis, MO, USA). Optical density of formazan crystals, formed by NAD(P)H-dependent oxidoreductases and dissolved in dimethyl sulfoxide, was measured after incubation for 2 hours at 37 °C using a microplate reader (EL 808, Biotek Instruments, Winooski, VT, USA). The results of the viability tests were normalized to untreated cells (set at 100% viability) and the half lethal concentration (LC50) was calculated by an exponential regression analysis using the formula y = m × e^(b × x) (y: TCM concentration, m: slope of the regression line, e: exponential, b: intersection with the X-axis, x at 50: LC50) [11,12].

#### 2.1.4. Cell Proliferation Behavior

For analyses of the proliferation behavior, L929 cells were seeded into 96-well plates (Greiner Bio-One GmbH, Frickenhausen, Germany) in growth medium at 30,000 cells/cm^2^ and stimulated the next day. For stimulation with the respective toothpastes, L929 cells were incubated with different concentrations of TCM for exactly two minutes, since two minutes is the recommended and also the average time for toothbrushing [14,15]. After washing the cells with PBS, serum-free media containing 5-bromo-2’-deoxyuridine (BrdU) was added for two hours according to the Cell Proliferation ELISA, BrdU (colorimetric) kit (Roche, Basel, Switzerland). The results were again normalized to untreated cells.

### 2.2. Statistical Analysis

Half lethal concentration values (LC50) of the toothpastes are reported (mean and standard deviation) from two independent experiments, each performed in triplicate. Differences in LC50 between cells treated with the five toothpastes were tested using a non-parametric Kruskal–Wallis test followed by a post-hoc Mann–Whitney U-test with Bonferroni correction for multiple comparisons (SPSS version 19.0, IBM, Armonk, NY, USA). Alpha of 5% was considered significant. The results on the cell proliferation behavior are described by the mean and standard deviation.

## 3. Results

### 3.1. Measurement of the pH Value and Free Fluoride

AirFlow was the only toothpaste showing a slightly basic pH value (7.75). The four commercially available toothpastes all showed a slightly acidic pH, with OB having the lowest value at 5.57. The SLS-free Ze also showed a low value of 6.08. The remaining two toothpastes, OM and Se, had nearly similar pH values of 6.75 and 6.69, respectively. Regarding the concentration of free fluoride, the SLS-free, commercially available toothpaste Ze had the highest value of 1575 ppm. The experimental AF toothpaste had 1400 ppm and the three remaining commercially available toothpastes had 1290 ppm (OM), 1106 ppm (Se), and 1050 ppm (OB) (Table 1).

### 3.2. Half Lethal Concentration (LC50)

The half lethal concentration (LC50) of toothpaste-conditioned medium (TCM) of the toothpastes (one experimental toothpaste, AirFlow AF, and four commercially available toothpastes, Zendium Ze, Sensodyne Se, OdolMed OM, and OralB OB) on L929 cells, gingival fibroblasts (GF), and periodontal ligament cells (PDL) are shown in Figure 1 (L929), Figure 2A (GF), and Figure 2B (PDL). The mean LC50 values of the toothpastes on L929 are 9.3 ± 4.6 for AF, 0.2 ± 0.3 for OB, 0.9 ± 0.9 for OM, 0.6 ± 0.6 for Se, and 5.6 ± 1.8 for Ze. AirFlow resulted in statistically significant higher LC50 values when compared with OB, OM, and Se (all *p* < 0.01). Only in comparison to the second SLS-free toothpaste (Zendium), AF showed no statistically significant difference in the LC50 values. Comparing the commercially available toothpastes only, Ze showed significantly higher LC50 values than OB and OM (both *p* = 0.03). Compared to Se, Ze also showed higher LC50 values, but without statistical significance after Bonferroni correction. The LC50 values of the toothpastes on the primary cells of the oral cavity, gingival fibroblasts (GF), and periodontal ligament cells (PDL) confirm the results obtained from the L929 cell tests.

### 3.3. Cell Proliferation Behavior

The proliferation behavior, investigated by the incorporation of BrdU into the DNA of L929 cells after stimulating the cells with different concentrations of TCM, of the five toothpastes mainly confirms the results of the viability testing. While cells returned to a proliferation rate of more than 80% when incubated with 6 *w*/*v*% AirFlow TCM, all commercially available toothpastes remained below a proliferation rate of 26%. When incubating the cells with 3% TCM, AF even resulted in unimpaired proliferation behavior when compared to the untreated control, while the toothpastes OM, OB, Se, and also Ze were still strongly impaired in their proliferation behavior (Table 2).

## 4. Discussion

In the present study, the chemical and biological parameters of a newly developed toothpaste were investigated and compared with four commercially available toothpastes. The main differences between the toothpastes, which all had a fluoride content above 1000 ppm, are, on the one hand, the addition of the possible caries-protective substances erythritol and xylitol to the newly developed toothpaste and, on the other hand, the use of the different surfactants. While three of the commercially available toothpastes contained sodium lauryl sulfate (SLS) as a surfactant, the commercially available toothpaste Zendium and the experimental toothpaste AirFlow used alternative surfactants.

The chemical parameters investigated included the measurement of pH and free fluoride concentration, while the biological parameters included the calculation of the half lethal concentration of the toothpastes in cell culturing tests and the investigation of the proliferation behavior of the cells. Remarkably, the newly developed toothpaste (AF) and the second SLS-free toothpaste (Ze) showed the best results in terms of compatibility for the cells investigated. Regarding the free fluoride content, AirFlow and Zendium, with a detected fluoride concentration of 1400 ppm and 1575 ppm, respectively, were in the upper range of the WHO recommendation or even above. OM, with a free fluoride content of 1050 ppm, was at the lower end of the recommendation, and toothpastes OB and Se, with free fluoride concentrations of 1290 ppm and 1100 ppm, were in the middle range. However, the similarities between the two SLS-free toothpastes, AF and Ze, did not extend to the measured pH value. While AF exhibited the highest basic pH value of 7.75, Ze showed an acidic pH value of 6.08. While OM had the lowest pH value of 5.57, OB and Se showed similar values of 6.75 and 6.69 also for this parameter.

Regarding fluoride concentration, it must be noted that all toothpastes tested had proven fluoride concentrations above 1000 ppm. Although the caries-protective effect of the investigated toothpastes was not investigated in the present study, the protection against caries by the use of fluoride-containing toothpastes above a concentration of 1000 ppm has been confirmed by numerous studies and consequently by a systematic review by the Cochrane Collaboration [16]. Additionally confirmed is that the caries-protective effect increases with increasing fluoride concentration, which is why toothpastes for adults should have fluoride concentrations in the upper range of, but not above, the WHO recommendation.

Finally, the question arises as to the significance of the individual parameters for the clinic. Looking at the chemical parameters of the study, it can be seen that all except one toothpastes complied with the WHO recommendations regarding the fluoride concentration and the International Organization for Standardization (ISO) recommendations regarding pH. However, it should be noted that a new measurement method was described in the literature explaining higher pH values in the original toothpaste than in toothpaste slurries [17]. Whether this has an influence on the incorporation of fluoride ions as well as on better protection of the demineralized tooth structure, which occurs better at a lower fluoride concentration in a slightly acidic environment [18], must be investigated in clinical studies.

In addition to the actual requirements that toothpaste must fulfill, such as cleaning the teeth, releasing fluoride, and much more, toothpaste must also be well tolerated by the tissue surrounding the teeth. Regarding the biological parameters of the toothpastes studied, the calculation of the half lethal concentration (LC50 values) was performed on different kinds of cells. AirFlow showed significantly higher LC50 values, which indicated its lower toxicity compared to the three commercially available toothpastes (OB, OM, and Se). This confirms the good tolerance of the toothpaste at the cell biological level. The fourth commercially available toothpaste (Ze) also showed high LC50 values. One explanation for the better cell survival after stimulation with AF and Ze could be the presence of alternative surfactants than sodium lauryl sulfate, which has already been shown by other studies [11,12].

Regarding the proliferation behavior of the cells after incubation with the respective toothpastes, the results of the viability tests are confirmed insofar as the cells after incubation with AF showed higher proliferation behavior compared to the cells incubated with toothpastes containing SLS. Interestingly, cells after incubation with AF showed higher proliferation rates compared to cells treated with the SLS-free toothpaste, Ze. The toothpaste AF resulted in a cell proliferation rate of 87% while Ze, known for its good biocompatibility, resulted in 26% after incubation with 6 *w*/*v*% toothpaste slurry. This additional positive cell-protective function might be provided in AirFlow by the addition of erythritol. Erythritol was previously shown to protect endothelial cells under hyperglycemic conditions [19]. In addition, membrane-protective properties have also been demonstrated in vitro [20]. In the newly developed toothpaste, erythritol is added as an active agent. A caries-protective effect of erythritol is also described in numerous studies, which can be attributed to the following points: the growth, as well as the acid production of Streptococcus mutans, the lead germ for the development of caries, is inhibited; the adherence of bacteria to the tooth structure is impeded, as well as the biofilm production is reduced. The caries-preventive efficacy of erythritol is sometimes even reported to be higher than that of xylitol, another sugar alcohol [21]. Despite the described effectiveness in the fight against caries, xylitol and erythritol are mainly used as sugar substitutes and erythritol is not found in toothpastes currently. A Cochrane analysis from 2015 showed a positive effect of fluoride-containing toothpastes with added xylitol compared to fluoride-containing toothpastes without added xylitol in the prevention of caries in children. However, the positive conclusion was weakened by a high bias risk and thus required further studies [22]. Until now, no studies with erythritol-containing fluoride toothpaste have been published. However, a long-term study discovered a positive effect on caries prevention in children with mixed dentition even 3 years after regular consumption of candies containing erythritol in comparison to xylitol and sorbitol [23]. Another study with erythritol-containing candies showed that erythritol was associated with the lowest incidence of Steptococcus mutans and also the lowest prevalence of caries [24]. Since the regular consumption of candies, however, is not ideal from the point of view of health education, the addition of erythritol to toothpastes offers a promising solution. 

## 5. Conclusions

In conclusion, the newly developed erythritol- and xylitol- containing toothpaste AirFlow shows excellent biocompatibility in cell tests and has high availability of free fluoride. Further studies on other important points for a toothpaste in daily use, such as the effectiveness in removing plaque, the effect on bacteria, as well as the performance in a caries model, are of course necessary and in planning.

## Figures and Tables

**Figure 1 healthcare-09-00935-f001:**
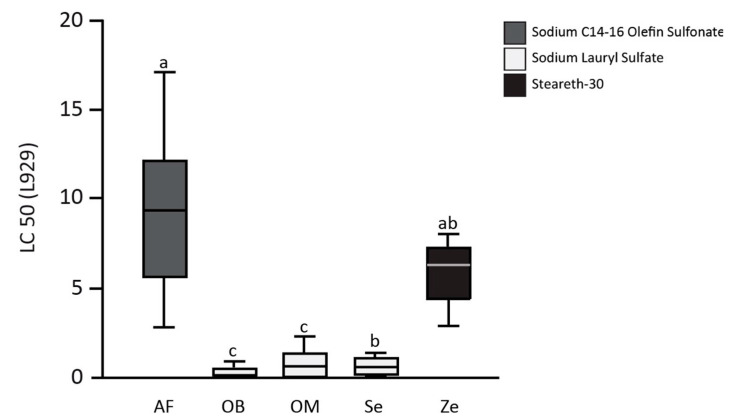
The half lethal concentration (LC50) of toothpaste-conditioned medium (TCM) of the toothpastes (one experimental toothpaste, AirFlow AF, and four commercially available toothpastes, Zendium Ze, Sensodyne Se, OdolMed OM, and OralB OB) on L929 cells. The different letters indicate statistically significant differences.

**Figure 2 healthcare-09-00935-f002:**
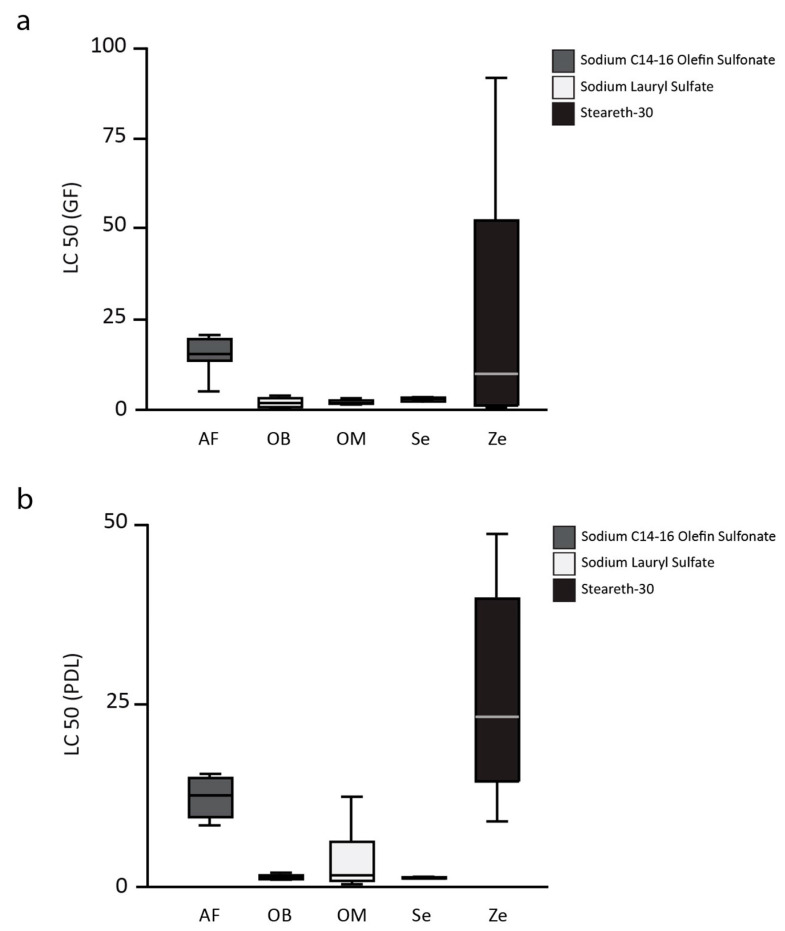
The half lethal concentration (LC50) of toothpaste-conditioned medium (TCM) of the toothpastes (one experimental toothpaste, AirFlow AF, and four commercially available toothpastes, Zendium Ze, Sensodyne Se, OdolMed OM, and OralB OB) on gingival fibroblasts (**a**) and periodontal ligament fibroblasts (**b**).

**Table 1 healthcare-09-00935-t001:** Toothpastes and the respective surfactants, amount of declared fluoride concentration, amount of free fluoride measured, and pH.

Toothpaste	Tenside (Surfactant)	Fluoride Declared	Free Fluoride Measured	pH
AF	Sodium C14-16 Olefin Sulfonate	1400 ppm NaF	1400 ppm	7.75
OM	Sodium Lauryl Sulfate	1450 ppm NaF	1290 ppm	6.75
OB	Sodium Lauryl Sulfate	1100 ppm SnF2 350 ppm NaF	1050 ppm	5.57
Se	Sodium Lauryl Sulfate	1100 ppm SnF2	1106 ppm	6.69
Ze	Steareth-30	1450 ppm NaF	1575 ppm	6.08

**Table 2 healthcare-09-00935-t002:** Proliferation behavior in % compared to untreated L929 fibroblasts after stimulation with different concentrations of toothpaste-conditioned medium (TCM).

TCM *w*/*v*%	AF	OM	OB	Se	Ze
25	25.9	9.0	10.7	11.7	19.4
12.5	55.9	14.8	11.9	12.7	18.5
6.3	86.8	19.1	13.2	14.7	25.4
3.1	100.4	31.1	17.3	13.8	35.7
1.6	103.7	46.4	23.0	17.1	70.2
0.8	101.0	74.1	54.1	37.0	78.0

## Data Availability

The datasets used and analyzed during the current study are available from the corresponding author on reasonable request.

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
