# Peer review of "The Biocompatibility of a New Erythritol-and Xyltol-Containing Fluoride Toothpaste"

_healthcare, 2021, doi:10.3390/healthcare9080935_

Round 1

Reviewer 1 Report

The present study investigated the LC50 and cell proliferation behavior of L929 cells, GF, and PDL treated with 5 different toothpastes. The followings are some specific comments.

  1. This study used primary cells (GF and PDL). Therefore, the case number of IRB approval and the name of institute that permitted should be provided.
  2. How were GF and PDL isolated from extracted wisdom teeth?
  3. The amount of L929 cells, GF, and PDL for LC50 was not written.
  4. The amount of L929 cells for cell proliferation behaviour was not written.
  5. Both LC50 and cell proliferation behavior tested 2 mins of incubation of cells with TCM. The rationale of selecting 2 mins should be described.

Author Response

Response to Reviewer 1

Point 1: This study used primary cells (GF and PDL). Therefore, the case number of IRB approval and the name of institute that permitted should be provided.

Response 1:

Thank you for the advice.

The case number is 123/2020 (ethics committee Sigmund Freud University, Vienna). We added the information about the ethical approval at the Materials and Methods section.

Point 2: How were GF and PDL isolated from extracted wisdom teeth?

Response 2: We now added this information to complete the materials and methods section:

Human gingiva and periodontal ligament fibroblasts were obtained from tissue grafts after approval (123/2020 ethics committee SFU) and after informed consent by the donor. To isolate the cells, tissue explants from the periodont and the gingiva of extracted wisdom teeth were cultivated in Dulbecco’s Modified Eagle Medium (DMEM, Invitrogen Corporation, Carlsbad, CA) supplemented with 10% fetal calf serum (FCS; PAA Laboratories, Linz, Austria) and antibiotics (Invitrogen) at 37°C, 5% CO2 and 95% humidity. Fibroblasts that grew out from the explants were used for the experiments up to the seventh passage.

Point 3: The amount of L929 cells, GF, and PDL for LC50 was not written.

Response 3: L929 cells, GF, and PDL were all seeded in growth medium at 30.000 cells/cm2 into culture dishes (Greiner Bio-One GmbH, Frickenhausen, Germany). For all experiments 96well plates were used.

Point 4: The amount of L929 cells for cell proliferation behaviour was not written.

Response 4: Also for proliferation assays, cells were seeded into 96well plates (Greiner Bio-One GmbH, Frickenhausen, Germany). Cells were seeded in growth medium at 30,000 cells/cm2 and stimulated the next day.

Point 5: Both LC50 and cell proliferation behavior tested 2 mins of incubation of cells with TCM. The rationale of selecting 2 mins should be described.

Response 5: Cells were incubated with the respective TCM for exactly 2 minutes since two minutes are recommended and also an average time for toothbrushing [14, 15]

14  Creeth, JE. Gallagher, A. Sowinski, J. Bowman, J. Barrett, K. Lowe, S. Patel, K. Bosma ML. The effect of brushing time and dentifrice on dental plaque removal in vivo. J Dent Hyg. 2009, 83(3), 111-116.

15  Winterfeld, T. Schlueter, N. Harnacke, D. Illig, J. Margraf-Stiksrud, J. Deinzer, R Ganss, C. Toothbrushing and flossing behaviour in young adults--a video observation. Clin Oral Investig. 2015, 19(4), 851-858.

Reviewer 2 Report

This project refers to the investigation of several biological parameters of a new toothpaste (containing fluoride and erythritol but no SLS) and their comparison to other commercially available toothpastes. The overall work is interesting. Some points to consider:

Lines 88 and 184: Should state the No of Table and preferably place the header over the table. Check the instructions for authors section in the journal homepage/word template file. Also tables should be placed in a single page. This could be applied to figures as well for aesthetic reasons.

Author Response

Response to Reviewer 2

Point 1: This project refers to the investigation of several biological parameters of a new toothpaste (containing fluoride and erythritol but no SLS) and their comparison to other commercially available toothpastes. The overall work is interesting. Some points to consider:

Lines 88 and 184: Should state the No of Table and preferably place the header over the table. Check the instructions for authors section in the journal homepage/word template file. Also tables should be placed in a single page. This could be applied to figures as well for aesthetic reasons.

Response 1: Thank you for the positive revision. We have now added the tables and figures to the template according to the author’s guidelines.

Round 2

Reviewer 1 Report

The manuscript can be accepted in the present form.

Author Response

Thank you very much for the new check and the positive evaluation.